# A universal Urbach rule for disordered organic semiconductors

Christina Kaiser [1], Oskar J. Sandberg [1✉], Nasim Zarrabi[1], Wei Li[1], Paul Meredith [1] & Ardalan Armin [1✉]

In crystalline semiconductors, absorption onset sharpness is characterized by temperature-dependent Urbach energies. These energies quantify the static, structural disorder causing localized exponential-tail states, and dynamic disorder from electron-phonon scattering. Applicability of this exponential-tail model to disordered solids has been long debated. Nonetheless, exponential fittings are routinely applied to sub-gap absorption analysis of organic semiconductors. Herein, we elucidate the sub-gap spectral line-shapes of organic semiconductors and their blends by temperature-dependent quantum efficiency measurements. We find that sub-gap absorption due to singlet excitons is universally dominated by thermal broadening at low photon energies and the associated Urbach energy equals the thermal energy, regardless of static disorder. This is consistent with absorptions obtained from a convolution of Gaussian density of excitonic states weighted by Boltzmann-like thermally activated optical transitions. A simple model is presented that explains absorption line-shapes of disordered systems, and we also provide a strategy to determine the excitonic disorder energy. Our findings elaborate the meaning of the Urbach energy in molecular solids and relate the photo-physics to static disorder, crucial for optimizing organic solar cells for which we present a revisited radiative open-circuit voltage limit.

[1] Sustainable Advanced Materials (Sêr-SAM), Department of Physics, Swansea University, Singleton Park, Swansea, UK. ✉email: o.j.sandberg@swansea.ac.uk; ardalan.armin@swansea.ac.uk

Recently, research on organic solar cells has seen significant progress through the development of non-fullerene electron acceptors (NFAs), in particular delivering increases in single-junction efficiencies from 13.1 to 18.7 % between 2017 to 2021[1,2]. This has revived ambitions to ultimately achieve industrial scale low-cost photovoltaics with low embodied manufacturing energy. While chemists have been productively synthesizing new materials, understanding of the opto-electronic properties lags behind – particularly in relation as to why these new NFA blends are so effective at photogeneration with low voltage losses. A particular area of intense interest is light absorption and how it is related to molecular dipole moments and their energetic disorder. Of course, this is also a very relevant question for the fundamental solid-state physics of molecular and disordered semiconductors. Furthermore, energetic disorder critically determines dominant radiative loss mechanisms limiting the open-circuit voltage in optoelectronic applications.

In general, semiconductors tend to absorb light at photon energies below the bandgap (sub-gap absorption) depending on the energetic disorder. In the case of inorganic semiconductors, the absorption coefficient ($\alpha$) often displays an exponential tail below the bands. These so-called Urbach tails increase their broadening with temperature $T$[3]. The sub-gap $\alpha$ generally follows the expression

$$\alpha(E, T) \propto \exp\left(\frac{E - E_{on}(T)}{E_U(T)}\right), \quad (1)$$

where $E$ is the photon energy, $E_{on}$ is the energy onset of the tail, and $E_U(T)$ is the Urbach energy quantifying the total energetic disorder of the system. Depending on the semiconductor, $E_U$ generally varies between 10 to 100 meV at room temperature[4–7]. In banded semiconductors, it has been suggested that $E_U(T) = E_{U,D}(T) + E_{U,S}$, where $E_{U,D}(T)$ is a temperature-dependent dynamical disorder term related to the thermal occupation of phonon states[4,8], while $E_{U,S}$ is the width of the assumed exponential distribution of sub-gap states induced by static disorder[9]. However, a unifying theory describing the density of states and their absorption leading to Urbach tails for materials of different chemical bonding and morphology is still lacking[10–12].

In non-excitonic amorphous semiconductors, the definition of a clear bandgap edge is often difficult due to large static disorder inducing sub-gap broadening. Organic semiconductors, which are excitonic and partially amorphous, display even more complex sub-gap features including intermolecular hybrid charge transfer (CT) states in technologically-relevant blends of electron donors (D) and acceptors (A), excitonic features[13] and trap states[14]. CT states typically give rise to light absorption with Gaussian sub-gap spectral line-shapes. This has been attributed to intermolecular D:A transitions described by non-adiabatic Marcus theory[15] or its extensions[16–19], suggesting that the static disorder is governed by a Gaussian distribution of CT states. Whether a similar description applies for sub-gap absorption by intramolecular excitons is unclear. With the rise of NFA semiconductors with small energetic offset relative to donors[20], the lack of CT state sub-gap spectral features in donor-acceptor blends has led to the increased usage of Urbach energies to understand disorder and sub-gap photo-physics. As a rule of thumb, it has been assumed that low Urbach energies are indicative of lower static disorder and hence expected to result in higher performance such as reduced charge recombination and higher charge carrier mobilities in a photovoltaic device[21–23]. Due to the complex and often convoluted spectral features in sub-gap light absorption of organic semiconductors, however, the Urbach energy carries significant ambiguity. Moreover, a long-standing debate[24–26] on the distribution of density of states (DOS) defining the static disorder in organic semiconductors has been revived: does the DOS follow a Gaussian or an exponential distribution? With no doubt, the origin of the static disorder is of great importance for the classification and future development of organic semiconductors and their electro-optical properties. However, it has remained unclear how this important Figure-of-merit relates to the Urbach energy.

In this work, we show that the exciton sub-gap absorption in organic semiconductors, at photon energies well below the gap, is generally characterized by Urbach tails with characteristic energies equivalent to the thermal energy $kT$ as demonstrated by temperature-dependent external quantum efficiency (EQE) measurements ($k$ is the Boltzmann constant). However, these Urbach tails are often convoluted with Gaussian line-shapes induced by trap states and/or CT states, resulting in erroneous, energy-dependent, Urbach energies larger than $kT$. A simple model, combining the Gaussian distributed DOS with Boltzmann-like thermally activated optical transitions, is shown to reproduce the absorption profiles and to give an estimate for the Gaussian static disorder. Based on this model, the exciton sub-gap absorption is found to be composed of two regimes: near the onset, the sub-gap absorption is dominated by Gaussian static disorder resulting in strongly energy dependent Urbach energies, while thermal broadening dominates at energies well below the gap, where the Urbach energy approaches $kT$. As such, using an (energy-independent) Urbach energy as a probe for the static disorder in organic solar cells is meaningless. Finally, the effect of exciton static disorder and thermal broadening on the expected radiative voltage losses and power conversion efficiency (PCE) in organic solar cells based on low offset D:A blends is clarified.

## Results

The spectral line-shape of $\alpha$ and the absorptance $A$ in the sub-gap tail are generally related via a modified Beer-Lambert law, $A = \tilde{f} \alpha d$, where $d$ is the thickness of the active layer and $\tilde{f}$ is an energy-dependent correction factor accounting for optical interference[27,28]. For optically thin films with layer thicknesses of 100 to 150 nm, $\tilde{f}$ is often assumed to be close to 1 (negligible optical interference effects). Moreover, it has been shown for efficient D:A systems that the internal quantum efficiency (IQE) is generally excitation energy independent, hence EQE $\propto A$ and the spectral line shape of the EQE follows $\alpha$[29–32]. Based on this underlying premise, EQE measurements have been frequently employed in the past to determine $E_U$. However, a previous lack of sensitivity in the EQE measurements has led to speculative assumptions about the spectral range of trap state absorptions, exponential tails and associated $E_U$ in organic semiconductors. By choosing a small fitting range, exponential fits can be forced on to EQE spectra resulting in a rather arbitrary $E_U$ dependent on the spectral range of the fitting. More insight can instead be gained from the apparent Urbach energy ($E_U^{app}$) here defined as:

$$E_U^{app}(E) = \left[\frac{d\ln(EQE)}{dE}\right]^{-1} \quad (2)$$

For a true exponential tail in the form of Eq. 1, $E_U^{app}$ is constant in the sub-gap spectral region and given by $E_U$.

## Determination of the apparent Urbach energy in organic solar cells. Figure 1 shows the EQE and $E_U^{app}$ for a wide range of organic semiconductor D:A blends. The material systems studied here can be generally grouped according to the D:A energy offset or, in other words, the difference between the CT state energy $E_{CT}$ and the optical bandgap $E_{opt}$. Here, $E_{opt}$ is typically equal to the local exciton (LE) energy of the lower-bandgap component (between D and A). Large offset systems, such as PCDTBT:PC$_{70}$BM, BQR: PC$_{70}$BM and PBDB-T:PC$_{70}$BM (Fig. 1a), show three distinct

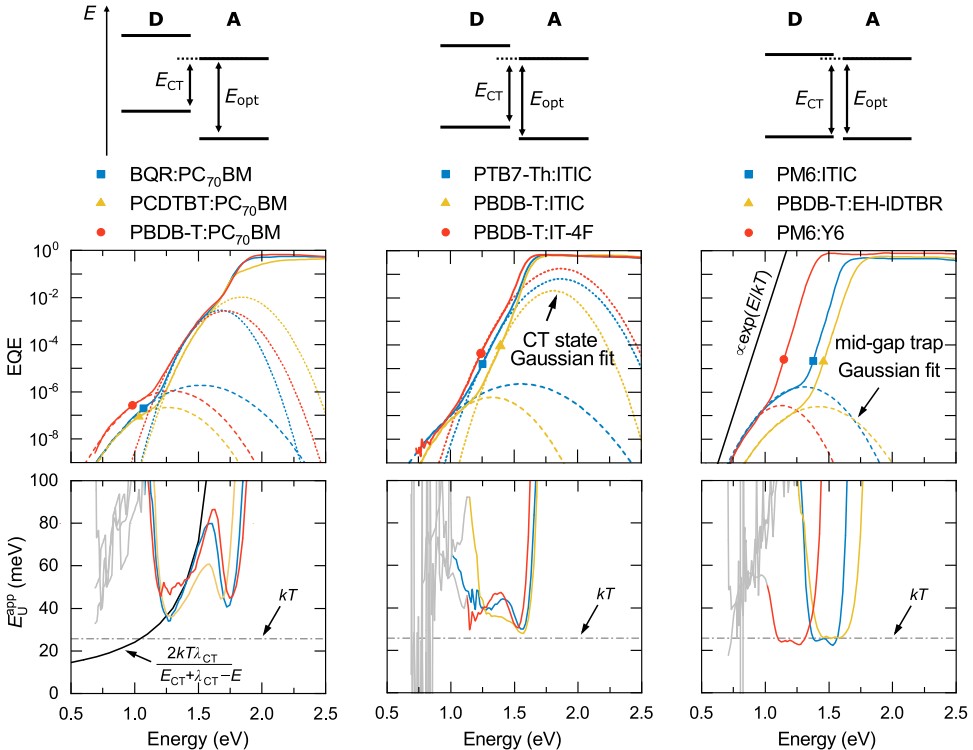

**Fig. 1 EQE and $E_U^{app}$ for organic D:A blends with different energetic offsets between $E_{CT}$ and $E_{opt}$.** Schematic illustration of the energetic offset ($E_{opt} - E_{CT}$) decreasing from left to right. Gaussian fits in the spectral range of CT state and trap state absorption are performed where possible. For low offset blends PM6:Y6, PM6:ITIC and PBDB-T:EH-IDTBR, the EQE tail has the form $e^{E/kT}$, and $E_U^{app}$ roughly equals $kT$ in the spectral range that is dominated by the absorption of the acceptor. For large offset material systems, such as BQR:PC$_{70}$BM, PBDB-T:PC$_{70}$BM and PCDTBT:PC$_{70}$BM, $E_U^{app}$ shows a $2kT\lambda_{CT}/(E_{CT} + \lambda_{CT} - E)$ dependence in the spectral range of CT absorption.

spectral ranges for energies below the gap: (i) mid-gap trap state absorption at energies below 1.2 eV, (ii) CT state absorption in the mid-range (~1.2–1.5 eV) and (iii) LE absorption above 1.5 eV. For mid-gap and CT state absorption, Gaussian-shaped EQE features are observed[14]. These features are consistent with Marcus charge-transfer between states that are distributed in accordance with a Gaussian density of sub-gap states. Details of the Gaussian fits are provided in the Methods section. Because of the Gaussian line-shape (see Equation 5 in the Methods), the corresponding $E_U^{app}$ in the spectral range of CT absorption is expected to be strongly energy-dependent and given by $E_U^{app} \approx 2kT\lambda_{CT}/(E_{CT} + \lambda_{CT} - E)$; this is highlighted by the black solid line in Fig. 1a. Here, $E_{CT}$ and $\lambda_{CT}$, as obtained from the Gaussian fits, are to be considered the effective energy and reorganization energy of CT states which includes the effect of static disorder (see Supplementary Note 2)[16,18]. At higher energies corresponding to the sub-gap LE absorption regime, in turn, $E_U^{app}$ has a narrow parabolic shape with a sharp minimum at roughly 40 − 50 meV for the large offset blends.

The EQE and $E_U^{app}$ of blends with a smaller offset between $E_{CT}$ and $E_{opt}$ (PTB7-Th:ITIC, PBDB-T:ITIC and PBDB-T:IT-4F) are shown in the Fig. 1b. In this case, the Gaussian CT state line-shape is barely recognizable, and the $\alpha$ tail in the spectral range of LEs is visible over a wider spectral range. While $E_U^{app}$ retains its parabolic shape in the LE sub-gap absorption regime, because of the wider range, the minimum value of $E_U^{app}$ is significantly reduced to values close to $kT$. Finally, the EQE and $E_U^{app}$ of blends in low-offset D:NFA systems (PM6:Y6, PM6:ITIC and PBDB-T: EH-IDTBR) are shown in Fig. 1c. In these blends, the CT state absorption can no longer be discerned from the EQE spectra. Instead, the $\alpha$ tail in the spectral range of LEs remains dominant

down to the energy range of mid-gap state excitation. In this limit, the $E_U^{app}$ in the LE-dominated sub-gap absorption range finally saturates and reaches a broad plateau where $E_U^{app} \approx kT$.

The above experimental observation for low offset D:NFA systems suggests the presence of an intramolecular sub-gap absorption regime where $\alpha \propto \exp(E/kT)$, but that its spectroscopic observation is obstructed by CT absorption in systems with larger offsets. To further clarify this, neat D or A systems (lacking the D:A CT absorption feature in the sub-gap EQE spectrum) were investigated as shown in Fig. 2. In the case of neat PC$_{70}$BM (Fig. 2a), a narrow spectral range where $E_U^{app} \approx kT$ can be identified. We note that this is consistent with photothermal deflection spectroscopy results[33] of neat PC$_{60}$BM (see Supplementary Fig. 3), confirming the underlying assumption that the spectral line shape of the EQE follows $\alpha$ in the sub-gap region. In PC$_{70}$BM, the spectral range where $E_U^{app} \approx kT$ is limited by deep trap-state absorption at low energies and low signal-to-noise ratio (SNR) due to the poor exciton dissociation in the neat phase (translating into a low Internal Quantum Efficiency [IQE]). However, adding 0.1 mol% of the wide-gap donor m-MTDATA results in enhanced exciton dissociation, which increases the spectral range within which $E_U^{app} \approx kT$ can be observed. Since m-MTDATA:PC$_{70}$BM is characterized by a low $E_{CT}$, the LE tail of PC$_{70}$BM can be clearly distinguished from CT states as shown in Fig. 2a[34]. By further increasing the donor content, the CT state absorption increases and the parabolic shape of $E_U^{app}$ emerges. This explains the parabolic shapes seen for large offset systems in Fig. 1a, which appear when the $\alpha$ tail in the spectral range of LE absorption becomes convoluted with CT state absorption. This is further supported by results shown in Fig. 2b for neat NFA devices comprising ITIC and IT-4F active layers, respectively.

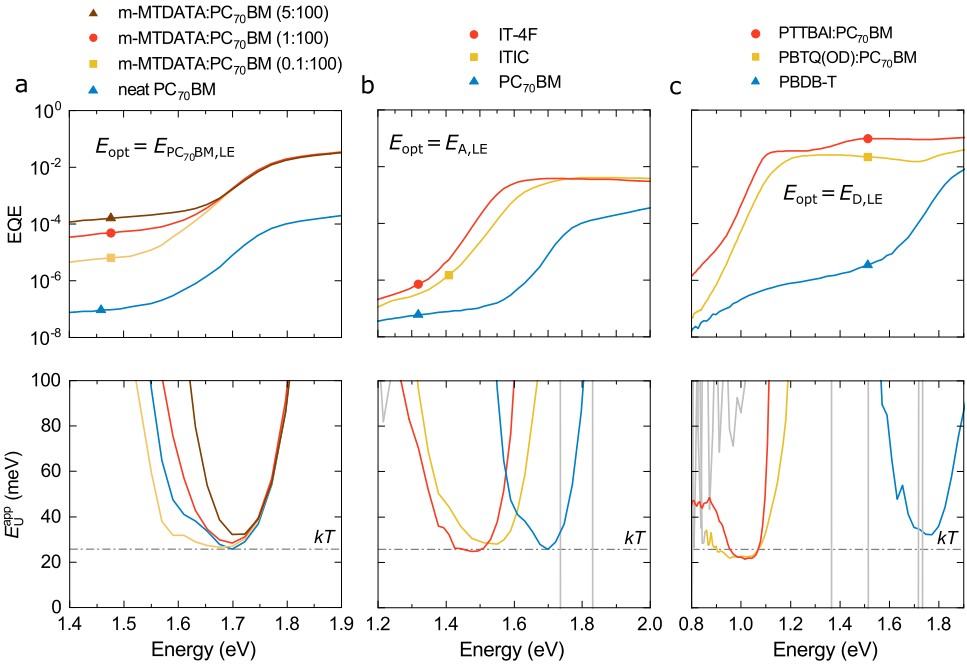

**Fig. 2 EQE tail dominated by neat material absorption. a**, EQE and $E_U^{app}$ for a series of m-MTDATA sensitized PC$_{70}$BM solar cells. The widest energy range over which $E_U^{app}$ equals $kT$ is observed for 0.1 mol% m-MTDATA in PC$_{70}$BM due to improved exciton dissociation in comparison to neat PC$_{70}$BM. Examples for EQE tails partially dominated by **b**, the neat donor absorption or by **c**, the neat acceptor absorption hence showing $E_U^{app} \approx kT$ in some parts of the spectrum. EQE spectra of PTTBAI:PC$_{70}$BM and PBTQ(OD):PC$_{70}$BM are taken from reference[35].

Here again, exponentially decaying $\alpha$ tail regions with $E_U^{app} \approx kT$ are observed.

To check if $E_U^{app} \approx kT$ is also true for donor materials has proven more challenging, since the neat donor absorption is often convoluted with deep trap-state absorption. This convolution effectively increases $E_U^{app}$ above $kT$ as shown for PBDB-T in Fig. 2c, where $E_U^{app} \approx 30$ meV. The problem can be circumvented by using narrow-gap polymer donors such as PBTQ(OD) or PTTBAI, which show strongly redshifted EQE spectra when blended with PC$_{70}$BM as previously reported[35]. As shown in Fig. 2c, $E_U^{app}$ is around 22 meV in this case.

While deviations of the type $E_U^{app} > kT + 10$ meV can be attributed to the presence of other absorbing species like CT states or deep trap states, small deviations of $E_U^{app}$ around $kT$ are likely caused by interference effects, i.e. non-constant $\widetilde{f}$, as shown previously[27,36]. Interference effects arise from the spectral dependence of the optical constants of the different layers in the thin-film stack and generally vary with the thickness of the active layer. To demonstrate the presence of interference effects, we therefore fabricated PM6:ITIC devices and PM6:Y6 devices of different active layer thicknesses. Indeed, as illustrated in the Supplementary Fig. 4, $E_U^{app}$ shows a weak thickness dependence leading to deviations around $kT$ from +2.3 meV to −5.8 meV. This is further corroborated by optical transfer-matrix simulations (see Supplementary Fig. 5).

**Temperature-dependent EQE measurements**. To verify that the exponential $\alpha$ tails in the spectral range of LEs are characterized by $E_U^{app} \approx kT$, we performed $T$-dependent EQE measurements on PBDB-T:EH-IDTBR, PM6:Y6 and PM6:ITIC, and the neat materials IT-4F and Y6. Figure 3a shows the normalized EQE and the respective $E_U^{app}$ spectra of three representative systems: PBDB-T:EH-IDTBR, PM6:Y6 and IT-4F. The remaining material systems are provided in the Supplementary Note 8. Depending on $T$, two different regimes can be distinguished in the sub-gap

absorption tail: At high $T$, $E_U^{app}$ spectra show a $T$-dependent plateau only influenced by interference effects and the shift in the absorption onset. At low $T$, however, the $E_U^{app}$ spectra generally attain a parabolic shape suggesting that the LE tail becomes convoluted with CT and/or mid-gap states. The thermal activation of the exponential $\alpha$ tail is then finally illustrated in Fig. 3b where $E_U^{app}$ at a constant energy in the plateau region is shown as a function of $kT$ for the systems studied. We see that $E_U^{app}$ at higher temperature is linear and equals $kT \pm 2.5$ meV, where the offset arises from interference effects. At lower $T$, $E_U^{app}$ eventually deviates from linearity, as the spectral shape is increasingly affected by other absorbing species. For example, the low-energy tail of the CT absorption may emerge at low $T$ owing to its distinctly different $T$-dependence compared to LE absorption[37]. Moreover, the absorption of trap states is expected to play a role as well, while little is yet known about their spectral broadening as a function of $T$.

For comparison, we also measured $T$-dependent EQE spectra of a commercial a-Si:H thin-film solar cell (EQE and $E_U^{app}$ spectra shown in Supplementary Fig. 7). In banded semiconductors such as a-Si:H, the total energetic disorder is $E_U(T) = E_{U,D}(T) + E_{U,S}$, consistent with the presence of an exponential DOS of tail states (defined by the associated width $E_{U,S}$). In Fig. 3b, two regimes can be distinguished for a-Si:H comprising the low-temperature saturation of $E_U$ to $E_U(0) \approx 40$ meV and thermal activation at higher temperatures with an offset of roughly 21 meV from $kT$ in agreement with previous reports[38–40]. In contrast, for the organic semiconductor systems in Fig. 3b, an extrapolation to $T = 0$ implies that $E_{U,S} \approx 0$, suggesting the lack of an exponential tail state distribution in these systems. These observations raise the question: what is the role of static disorder in shaping the $E_U^{app}$ spectra?

**Model for understanding the sub-gap $\alpha$**. In the context of the Marcus formalism for non-adiabatic charge transfer (high-

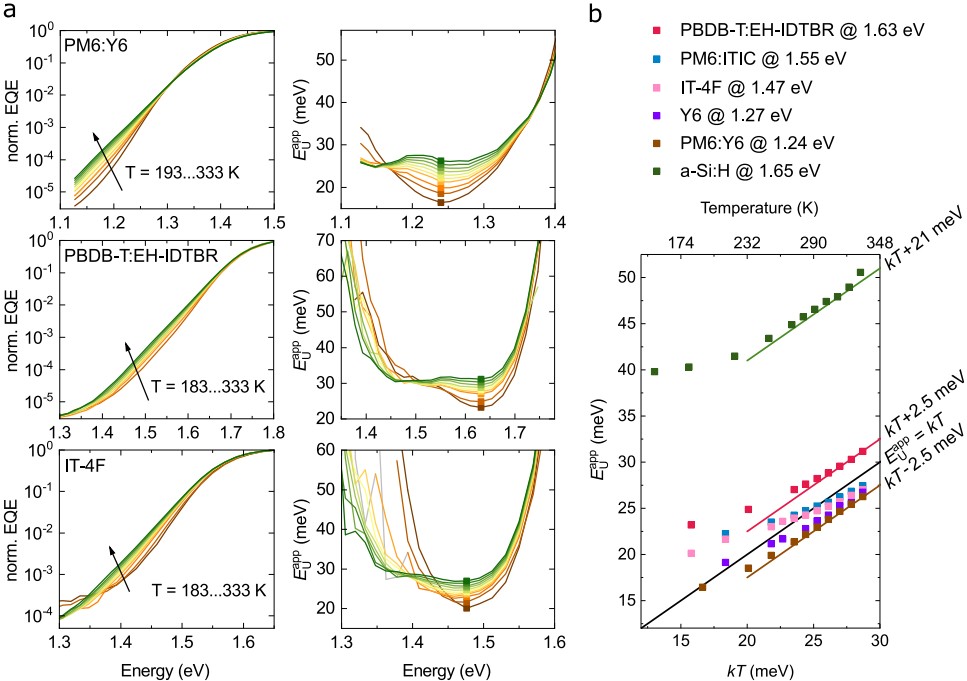

**Fig. 3 Temperature dependent apparent Urbach energies. a,** Normalized sub-gap EQE and $E_U^{app}$ spectra of inverted PM6:Y6, PBDB-T:EH-IDTBR and neat IT-4F solar cells at different temperatures. **b,** $E_U^{app}$ as a function of $kT$ for all tested materials systems including a commercial a-Si:H thin-film solar cell.

temperature limit), the presence of $E_U^{app} \approx kT$ in the spectral range of LE absorption (described by the absorption coefficient $\alpha_{LE}$) may be rationalized in terms of non-equal potential energy surfaces for the excited state and the ground state. By assuming a significantly more diffuse or delocalized excited state, compared to the strongly localized ground state, a much smaller reorganization energy is expected for the excited state in comparison to the ground state[41–43]. In this limit, we expect $\alpha_{LE}(E) \approx \alpha_{sat} \exp\left(\left[E - E_{opt}\right]/kT\right)$ for $E < E_{opt}$, in the case of negligible static disorder (see Supplementary Note 4). Here, $\alpha_{sat}$ includes a $1/E$ dependence, however, the sub-gap spectral lineshape is dominated by the Boltzmann factor. For above-gap absorption $E > E_{opt}$, we assume $\alpha_{LE}(E) \approx \alpha_{sat}$. We note that this simplification of $\alpha_{LE}(E)$ essentially follows a Miller-Abrahams-type charge-transfer formalism, sometimes used to describe exciton migration[44], but more commonly used for charge transport[45] in organic semiconductors. Accounting for static disorder described by a DOS given by $g_{DOS}\left(E'_{opt}\right)$, the total absorption coefficient is of the form $\alpha_{LE}(E) = \int \alpha_{LE}\left(E, E'_{opt}\right) g_{DOS}\left(E'_{opt}\right) dE'_{opt}$. In the Supplementary Note 3, expressions for the associated absorption coefficients are derived for the cases of a Gaussian DOS and an exponential tail DOS. For the case of an exponential tail DOS with the width $W$, we obtain $E_U^{app} = W$ (see Supplementary Fig. 8). This is indeed not what we observe in experiments. For a Gaussian DOS, in turn, we obtain $\alpha$ in the form

$$\frac{\alpha_{LE}(E)}{\alpha_{sat}} = \frac{1}{2}\left\{ \exp\left(\frac{E - E_{opt} + \frac{\sigma_s^2}{2kT}}{kT}\right) \left[1 - \text{erf}\left(\frac{E - E_{opt} + \frac{\sigma_s^2}{kT}}{\sigma_s\sqrt{2}}\right)\right] + \text{erf}\left(\frac{E - E_{opt}}{\sigma_s\sqrt{2}}\right) + 1 \right\}$$

(3)

where $\sigma_s$ is the standard deviation of the Gaussian DOS and $E_{opt}$ is the associated mean exciton energy. Here, $E_{opt}$ corresponds to the optical gap and is determined by the first excited singlet state of either

donor ($E_{opt} = E_{D,LE}$) or acceptor ($E_{opt} = E_{A,LE}$) in a blend. For $E \ll E_{opt}$, Eq. (3) reduces to the exponential part and $E_U^{app}$ therefore equals $kT$. On the other hand, the error functions govern the spectral shape near the absorption edge as demonstrated in Supplementary Fig. 9.

To validate the model, we applied Eq. (3) to $T$-dependent EQE spectra, as shown in Fig. 4a for neat Y6 and IT-4F. Using neat materials avoids the influences of CT states in the low-energy tail although trap states are always present at lower energies. Doing so, we obtain $\sigma_s$ values of $47.0 \pm 0.7$ meV for Y6 and $35.0 \pm 2.6$ meV for IT-4F (see Supplementary Fig. 11). More fittings on room temperature EQE spectra of other material systems are shown in the Supplementary Fig. 12). For blend systems, the full experimental sub-gap EQE can be reconstructed assuming $\alpha(E) = \alpha_{LE}(E) + \alpha_{CT}(E) + \alpha_t(E)$ constituting the exponentially decaying $\alpha_{LE}$ according to Eq. (3) and the sum of two Gaussian functions (CT states and deep trap states; see Eq. 5). In Fig. 4b, we demonstrate the model for PBDB-T:PC$_{70}$BM using the parameters $E_{opt} = 1.89$ eV and $\sigma_s = 60$ meV, and other Gaussian fit parameters summarized in Supplementary Table 1. PBDB-T:PC$_{70}$BM belongs to the group of blends for which $E_{CT} \ll E_{opt}$ with $E_{CT} - E_{opt} \approx 0.45$ eV. Nevertheless, the simplified model reproduces the strong spectral dependence of the experimental $E_U^{app}$, including the parabolic behavior in the LE absorption dominated spectral range. On the other hand, by keeping all other parameters constant while increasing $E_{CT}$ with respect to $E_{opt}$, the emergence of the $E_U^{app} = kT$ plateau at around $1.6 \pm 0.1$ eV can be reproduced.

In the absence of CT states, the above model predicts three sub-gap regimes of the $\alpha(E)$ based on the dominant disorder mechanism as illustrated in Fig. 4c. For photon energies close to the gap ($E > E_{opt} - \sigma_s^2/kT$), $E_U^{app}$ is dominated by Gaussian static disorder. In this regime, the absorption resembles a Gaussian-like shape, where the steepness close to the gap is determined by $\sigma_s$, causing a redshift of the effective energy gap with increasing $\sigma_s$. While $E_U^{app}$ does not directly represent $\sigma_s$ at any energy, a positive

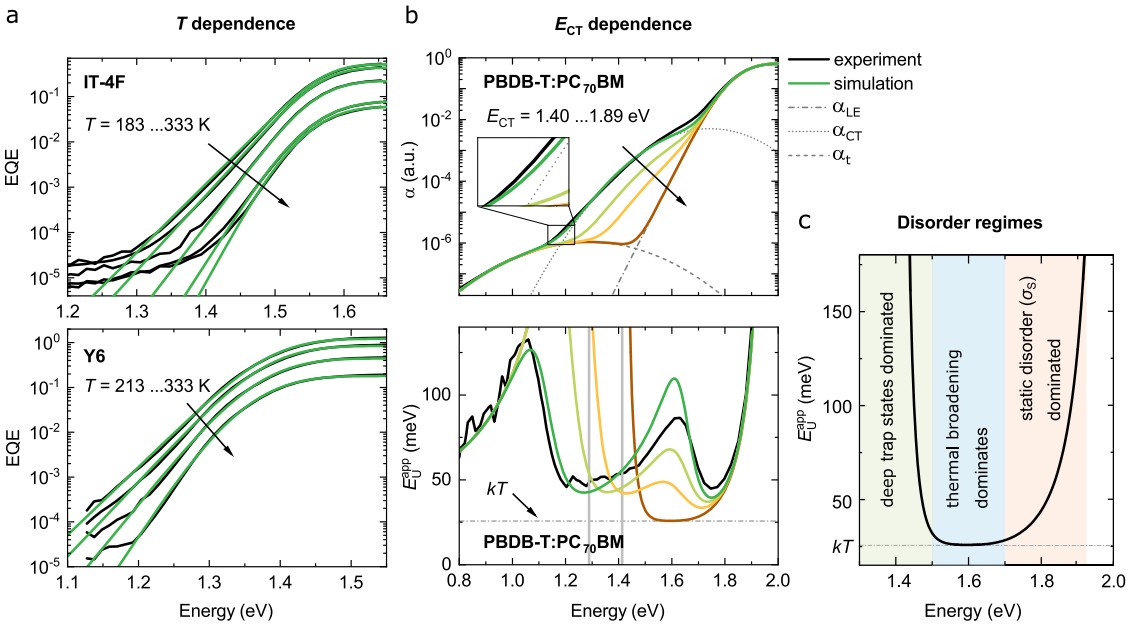

**Fig. 4 The sub-bandgap EQE as a function of $T$ and $E_{CT}$. a**, Experimental (black lines) and simulated (green lines) sub-gap EQE spectra of IT-4F and Y6. **b**, The line-shape of the experimental sub-gap EQE of PBDB-T:PC$_{70}$BM can be described as the sum of LE absorption, CT state absorption, and trap state absorption: $\alpha = \alpha_{LE} + \alpha_{CT} + \alpha_t$, with $\alpha_{LE}$ given by Eq. 3 ($E_{opt} = 1.89$ eV, $\sigma_s = 60$ meV), while $\alpha_{CT}$ and $\alpha_t$ are determined from their respective Gaussian fits. In the experiment, $E_U^{app}$ is well above $kT$, since $E_{opt} - E_{CT} > 0.3$ eV. Increasing $E_{CT}$ (see Eq. 5) in the simulation results in $E_U^{app} \to kT$ between 1.5 to 1.7 eV. **c**, Schematic representation of the disorder regimes dominating $\alpha$ in the absence of CT absorption comprising (i) static disorder $\sigma_s$ close to the band edge, (ii) thermal broadening, where $E_U^{app} = kT$, and (iii) deep trap state absorption well below the gap.

correlation between $E_U^{app}$ and $\sigma_s$ is observed in this regime (see Supplementary Fig. 10). A closer inspection suggests that $E_U^{app}$ is strongly energy dependent in this regime, scaling with $\sigma_s$ as $E_U^{app}(E) \approx 2\sigma_s^2/(E_X - E)$, for $\sigma_s < 4kT$, where $E_X$ is a constant independent of $E$. This explains the previously reported correlation between morphological disorder and $E_U^{app}$ close to the absorption onset[22,46–48]. For $E < E_{opt} - \sigma_s^2/kT$, however, the exponential term in Eq. (3) eventually starts to dominate the spectral line shape of $\alpha_{LE}$. In this regime, the sub-gap absorption of LEs is dominated by thermal broadening with $E_U^{app} \to kT$, thus becoming independent of $\sigma_s$. The transition between the static disorder dominated and the thermal broadening dominated regime depends on $\sigma_s$ (which is a measure of the Gaussian static disorder) and the temperature. This is simulated for the two cases $\sigma_s = 70$ meV and $\sigma_s = 100$ meV as shown in Supplementary Fig. 10: For $\sigma_s = 70$ meV, $E_U^{app}$ reaches $kT$ at $\alpha(E)$ values 3 to 4 orders of magnitudes below $\alpha_{sat}$, while this is 6 orders below $\alpha_{sat}$ for $\sigma_s = 100$ meV at room temperature. Finally, at energies well below the optical gap, $\alpha(E)$ eventually becomes dominated by deep trap state absorption, resulting in an artificial increase in $E_U^{app}$ and a concomitant deviation from $E_U^{app} = kT$. Deep trap state absorption is typically observed 6 orders of magnitude below $\alpha_{sat}$ limiting the spectral range dominated by thermal broadening.

Based on these considerations, we estimate that it is possible to observe $E_U^{app} \approx kT$ only when $\sigma_s < 100$ meV (see Supplementary Fig. 10). Other conditions that must be met are: (i) the neat phase absorption of one component is spectrally separated from the other neat phase absorption, as well as from the CT states and trap states; (ii) the dynamic range of the EQE (or $\alpha$) measurement is sufficiently wide to measure photocurrent at wavelengths well below the absorption onset of the neat material ($E < E_{opt} - \sigma_s^2/kT$); and (iii) optical cavity effects are not significant. Exponential sub-gap EQE spectra previously reported in literature for BHJs often do not fulfill these requirements, explaining reported Urbach energies much larger than $kT$.

Importantly, the $E_U^{app}$ spectra introduced here do not suffer from the short fitting ranges of previous of $E_U$ measurements. In contrast, we have shown that an exponential distribution of tail states cannot explain the sub-gap spectral line-shape associated with singlet absorption, nor CT or trap state absorption in organic semiconductors.

**Recombination losses and the radiative $V_{OC}$ limit set by excitons.** The presence of sub-gap absorption is known to induce radiative recombination losses additional to those predicted by the Shockley and Queisser (SQ) model. While the short-circuit current density ($J_{SC}$) remains largely unaffected, the presence of sub-gap absorption mainly translates into enhanced losses in the open-circuit voltage ($V_{OC}$). In general, the $V_{OC}$ can be expressed as $V_{OC} = V_{OC}^{SQ} - \Delta V_{OC}^{RAD} - \Delta V_{OC}^{NR}$, where $V_{OC}^{SQ}$ represents the upper thermodynamic limit of the $V_{OC}$ based on the SQ model, assuming perfect above-gap absorption and no sub-gap absorption. Furthermore, $\Delta V_{OC}^{RAD} = V_{OC}^{SQ} - V_{OC}^{RAD}$ is the radiative loss induced by sub-gap absorption, while $\Delta V_{OC}^{NR} = V_{OC}^{RAD} - V_{OC}$ is the non-radiative loss. Here, $V_{OC}^{RAD}$ is the radiative limit of the $V_{OC}$ corresponding to the expected $V_{OC}$ in the absence of non-radiative losses. In accordance with detailed balance, we expect $V_{OC}^{RAD} = \frac{kT}{q}\ln\left(\frac{J_{SC}}{J_0^{RAD}} + 1\right)$, where $J_{SC} = q\int_0^\infty \text{EQE}(E)\Phi_{sun}(E)dE$ and $J_0^{RAD} = q\int_0^\infty \text{EQE}(E)\Phi_{BB}(E)dE$ is the dark saturation current density in the radiative limit; $q$ is the elementary charge and $\Phi_{sun}$ ($\Phi_{BB}$) the solar spectrum (black body spectrum)[49]. The SQ limit ($V_{OC}^{RAD} = V_{OC}^{SQ}$) corresponds to the case when EQE = 1 for $E > E_{opt}$, while EQE = 0 for $E < E_{opt}$. We note that, because of the different ideality factors, the contribution from deep trap states will be negligible under 1-sun conditions, as shown previously[14].

As we have shown above, in low offset systems such as state-of-the-art NFA-based blends (or neat material systems), the CT absorption is mostly overshadowed by the stronger NFA absorption (or is absent). In such systems the sub-gap absorption

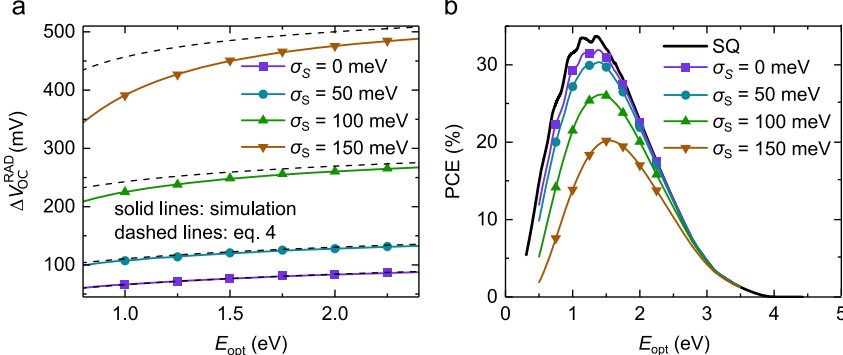

**Fig. 5 The radiative $V_{OC}$ loss induced by sub-gap absorption and associated radiative PCE limit of low-offset D:A solar cells. a**, The radiative voltage loss $\Delta V_{OC}^{RAD}$ as a function of the optical bandgap for varying degree of Gaussian static disorder $\sigma_s$, assuming the sub-gap absorption dominated by LEs as per Eq. 3 (solid lines with symbols). The corresponding analytical approximation (Eq. 4) is represented by the dashed lines. **b**, The corresponding radiative PCE limit based on Eq. 3, assuming EQE$_{max}$ = 1 and ideal charge collection, is shown (solid lines with symbols). For comparison, the SQ limit, representing the ideal case with no sub-gap absorption, is included as indicated by the black solid line. The PCE decreases with respect to the SQ limit as $\sigma_s$ increases. The SQ limit is not expected to be reached even for vanishing $\sigma_s$ due to the thermal broadening.

tail is dominated by excitons ($\alpha = \alpha_{LE}$). The expected radiative limit $V_{OC}^{RAD}$ set by excitons can be calculated by assuming EQE = EQE$_{max} \times (\alpha_{LE}/\alpha_{sat})$, with $\alpha_{LE}$ given by Eq. (3). The concomitant radiative loss $\Delta V_{OC}^{RAD}$, arising from the imperfect spectral line shape of $\alpha_{LE}$ (leading to deviations from the ideal box-like EQE profile), is shown in Fig. 5a at different $E_{opt}$ and $\sigma_s$. As shown, the voltage loss $\Delta V_{OC}^{RAD}$ increases with increasing $\sigma_s$. For example, when $\sigma_s$ = 100 meV, $\Delta V_{OC}^{RAD}$ is larger than 0.2 V. Furthermore, at large $\sigma_s$, the voltage loss becomes more prominent at larger $E_{opt}$. We note that these losses are caused by a drastic increase in $J_0^{RAD}$, while $J_{SC}$ is essentially unchanged. As $\sigma_s$ is reduced, in turn, $\Delta V_{OC}^{RAD}$ is correspondingly decreased. For small static disorder (i.e., small $\sigma_s$), an analytical approximation relating $\Delta V_{OC}^{RAD}$ and $\sigma_s$ can be obtained as

$$q\Delta V_{OC}^{RAD} \approx \frac{\sigma_s^2}{2kT} + kT \ln\left(\frac{E_{opt}}{3kT}\left[1 - \frac{\sigma_s^2}{2E_{opt}kT}\right]^3\right). \quad (4)$$

A good agreement between Eq. (4), as indicated by the dashed lines in Fig. 5a, and the full simulation is indeed obtained for small $\sigma_s$. Importantly, even in the limit of vanishing static disorder, $\sigma_s \to 0$, a voltage loss $\Delta V_{OC}^{RAD}$ of roughly 70–80 meV is expected due to non-vanishing sub-gap absorption induced by thermal broadening (manifested by $\alpha \propto \exp(E/kT)$ in the gap). Based on the $E_{opt}$ and $\sigma_s$ extracted (see Supplementary Fig. 12) for the low-offset D:A systems PM6:Y6 ($E_{opt}$ = 1.44 eV, $\sigma_s$ = 42 meV), PM6:ITIC ($E_{opt}$ = 1.69 eV, $\sigma_s$ = 37 meV) and PBDB-T: EH-IDTBR ($E_{opt}$ = 1.77 eV, $\sigma_s$ = 46 meV), the radiative voltage loss induced by sub-gap absorption can be estimated. Subsequently, we find $\Delta V_{OC}^{RAD}$ of 108 mV for PM6:Y6, 107 mV for PM6: ITIC, and 120 mV for PBDB-T:EH-IDTBR. We note that the corresponding would-be $\Delta V_{OC}^{RAD}$ (if CT states were absent, see Fig. 4b) for PBDB-T:PC$_{70}$BM is 150 mV.

Finally, Fig. 5b shows the corresponding effect of $\sigma_s$ on the radiative limit of the PCE, assuming EQE$_{max}$ = 1 and ideal charge collection. Compared to the SQ limit of the PCE (indicated by black solid line), the presence of the sub-gap absorption results in a PCE peak loss of around 1.5 % for $\sigma_s$ = 0. For $\sigma_s \neq 0$, the radiative PCE limit is further lowered, with the PCE peak decreasing with increasing $\sigma_s$. Hence, to minimize $\Delta V_{OC}^{RAD}$, and thus maximize PCE, it is important to minimize $\sigma_s$. However, an unavoidable radiative loss, relative to the SQ limit, will still be

present due to thermal broadening. Based on our findings, this loss is inherent to all organic solar cells, and needs to be taken into account in low offset systems where excitons dominate the sub-gap absorption.

## Discussion

In summary, we have shown that the exciton sub-gap absorption in organic semiconductors is dominated by Gaussian static disorder near the onset, while thermal broadening dominates at lower photon energies. Furthermore, we find that for a large number of organic semiconductors, the Urbach energy in the sub-gap absorption regime dominated by thermal broadening equals the thermal energy $kT$ within the variations caused by optical interference. While exponential absorption tails have been previously observed mainly in non-fullerene systems, we show that this property is universal for organic semiconductors and consistent with a Gaussian density of excitonic states undergoing Boltzmann-like thermally activated optical transitions. The static disorder is shown to arise from the width of the Gaussian DOS and to broaden the absorption onset at energies close to the optical gap. Using our model for the sub-gap excitonic absorption tail, it is possible to discriminate spectral regimes dominated by static disorder (where the Urbach energy is strongly dependent on the energy) and thermal broadening ($E_U = kT$), as well as to reproduce the temperature dependence of absorption coefficient due to excitons. This modified view of the sub-gap absorption coefficient in disordered organic semiconductors clarifies a longstanding debate concerning the shape of the DOS and the relevance of an Urbach description in these important and intriguing materials. Finally, we demonstrate the implications of exciton static disorder and thermal broadening on the radiative open-circuit voltage losses in organic solar cells based on low offset D:A blends.

## Methods

**Materials**. Poly(3,4-ethylenedioxythiophene) polystyrene sulfonate (PEDOT:PSS) was purchased from Heraeus. Zinc acetate dihydrate and PCDTBT (Poly[N-9′-heptadecanyl-2,7-carbazole-alt-5,5-(4′,7′-di-2-thienyl-2′,1′,3′-benzothiadiazole)]) were purchased from Sigma Aldrich. PC$_{70}$BM ([6,6]-Phenyl-C71-butyric acid methyl ester) and EH-IDTBR were purchased from Solarmer (Beijing). BQR (benzodithiophene-quaterthiophene-rhodanine) was provided by Prof. David. J Jones (University of Melbourne). m-MTDATA (4,4′,4''-Tris[(3-methylphenyl) phenylamino]triphenylamine) was purchased from Ossila. IT-4F (3,9-bis(2-methylene-((3-(1,1-dicyanomethylene)-6,7-difluoro)-indanone))-5,5,11,11-tetrakis (4-hexylphenyl)-dithieno[2,3-d:2′,3′-d']-s-indaceno[1,2-b:5,6-b']dithiophene), PM6 (Poly[(2,6-(4,8-bis(5-(2-ethylhexyl-3-fluoro)thiophen-2-yl)-benzo[1,2-b:4,5-b']dithiophene))-alt-(5,5-(1′,3′-di-2-thienyl-5′,7′-bis(2-ethylhexyl)benzo[1′,2′-

c:4′,5′-c']dithiophene-4,8-dione)]), Y6 (2,2′-((2Z,2′Z)-((12,13-bis(2-ethylhexyl)-3,9-diundecyl-12,13-dihydro-[1,2,5]thiadiazolo[3,4-e]thieno[2",3":4′,5′]thieno[2′,3′:4,5]pyrrolo[3,2-g]thieno[2′,3′:4,5]thieno[3,2-b]indole-2,10-diyl)bis(methanylylidene))bis(5,6-difluoro-3-oxo-2,3-dihydro-1H-indene-2,1-diylidene))dimalononitrile), ITIC (3,9-bis(2-methylene-(3-(1,1-dicyanomethylene)-indanone))-5,5,11,11-tetrakis(4-hexylphenyl)-dithieno[2,3-d:2′,3′-d']-s-indaceno[1,2-b:5,6-b'] dithiophene), PBDB-T (Poly[(2,6-(4,8-bis(5-(2-ethylhexyl)thiophen-2-yl)-benzo [1,2-b:4,5-b']dithiophene))-alt-(5,5-(1′,3′-di-2-thienyl-5′,7′-bis(2-ethylhexyl)benzo [1′,2′-c:4′,5′-c']dithiophene-4,8-dione)]) and PTB7-Th (Poly[4,8-bis(5-(2-ethylhexyl)thiophen-2-yl)benzo[1,2-b;4,5-b']dithiophene-2,6-diyl-alt-(4-(2-ethylhexyl)-3-fluorothieno[3,4-b]thiophene-)-2-carboxylate-2–6-diyl)]) were purchased from Zhi-yan (Nanjing) Inc.

**Device fabrication.** Solar cells were fabricated with either a conventional architecture Indium tin oxide (ITO)/PEDOT:PSS/active layer/Ca/Al or inverted architecture ITO/ZnO/active layer/MoO₃/Ag. Commercial ITO coated glass substrates from Ossila were cleaned in an aqueous solution of Alconox at 60 °C, followed by an ultrasonic bath in deionize water, acetone and isopropanol. The cleaned substrates were dried with nitrogen, followed by a UV/O₃ treatment (Ossila, L2002A2-UK). For the conventional device architecture, 30 nm of PEDOT:PSS was spin-coated at 6000 rpm for 30 s onto precleaned ITO substrates and annealed at 155 °C for 15 min. As top electrode, 20 nm of calcium (Ca) and 100 nm of Aluminum (Al) were vacuum deposited at $10^{-6}$ Tor defining an active area of 4 mm². For the inverted device architecture with 30 nm of ZnO, a solution of 200 mg of zinc acetate dihydrate in 2-methoxyethanol (2 ml) and ethanolamine (56 μl) was prepared and stirred overnight under ambient conditions. The ZnO layer formed upon spin-coating the solution at 4000 rpm followed by thermal annealing at 200 °C for 60 min. As top electrode, 7 nm of MoO₃ and 100 nm of Ag were vacuum deposited at $10^{-6}$ Tor defining an active area of 4 mm².

**Devices with conventional structure.** BQR:PC₇₀BM devices were prepared using with the conventional architecture ITO/PEDOT:PSS/BQR:PC₇₀BM/Ca/Al. BQR and PC₇₀BM were dissolved in toluene (24 mg/ml with the donor:acceptor ratio of 1:1) and stirred at 60 °C for 3 h. Next, the BQR:PC₇₀BM solution was spin-coated at 1000 rpm on the PEDOT:PSS layer to form a 100 nm thick film.

**Devices with inverted structure.** m-MTDATA:PC₇₀BM devices: Equimolar solutions of PC₇₀BM and m-MTDATA in dichloromethane (DCM) with a concentration of 19.4 mmol/l were prepared. To obtain a series of solutions with different molar ratios of m-MTDATA:PC₇₀BM (5 mol%, 1 mol%, 0.1 mol% and 0 mol% of m-MTDATA), 50 μl, 10 μl, 1 μl and 0 μl of m-MTDATA in DCM were added to 1 ml of PC₇₀BM in DCM. The solutions were spin-coated at a spin rate of 800 rpm to get an active layer thickness of around 90 nm. PM6:Y6 devices: PM6:Y6 was dissolved in chloroform (CF) solution (14 mg ml⁻¹ with 0.5 vol.% 1-Chloronaphthalene [CN]) with a donor:acceptor ratio of 1:1.2, and spin-coated (3000 rpm) on ZnO to form a 100 nm thick film. The as cast active layers were thermally annealed at 110 °C for 10 min. PM6:ITIC devices: PM6:ITIC was dissolved in chlorobenzene (CB) solution (18 mg ml⁻¹ with 0.5 vol.% DIO) with a donor:acceptor ratio of 1:1, and spin-coated (1000 rpm) on ZnO to form a 100 nm thick film. The active layers were further treated with thermal annealing at 100 °C for 10 min. PBDB-T:EH-IDTBR devices: PBDB-T:EH-IDTBR was dissolved in CB solution (14 mg ml⁻¹) with a donor:acceptor ratio of 1:1, and spin-coated (800 rpm) on ZnO to form a 100 nm thick film. PBDB-T:ITIC devices: PBDB-T:ITIC was dissolved in CB solution (14 mg ml⁻¹ with 0.5 vol.% DIO) with a donor:acceptor ratio of 1:1, and spin-coated (800 rpm) on ZnO to form a 100 nm thick film. The active layers were further treated with thermal annealing at 100 °C for 10 min. PTB7-Th:ITIC devices: PTB7-Th:ITIC was dissolved in CB solution (14 mg ml⁻¹ with 1 vol.% 1,8-diiodooctane [DIO]) with a donor:acceptor ratio of 1:1.4, and spin-coated (1000 rpm) on ZnO to form a 100 nm thick film. PBDB-T:PC₇₀BM devices: PBDB-T:PC₇₀BM was dissolved in CB solution (14 mg ml⁻¹ with 3 vol.% DIO) with a donor:acceptor ratio of 1:1.4, and spin-coated (1000 rpm) on ZnO to form a 100 nm thick film. Then the as-cast films were rinsed with 80 μL of methanol at 4000 rpm for 20 s to remove the residual DIO. PBDB-T:IT-4F devices: PBDB-T:IT-4F was dissolved in CB solution (14 mg ml⁻¹ with 0.5 vol.% DIO) with a donor:acceptor ratio of 1:1, and spin-coated (800 rpm) on ZnO to form a 100 nm thick film. The active layers were further treated with thermal annealing at 100 °C for 10 min. Neat ITIC devices: ITIC was dissolved in CF solution (10 mg ml⁻¹) and spin-coated on ZnO (2000 rpm) to form a 70 nm thick film. Neat IT-4F devices: IT-4F was dissolved in CF solution (10 mg ml⁻¹) and spin-coated on ZnO (2000 rpm) to form a 70 nm thick film. Neat PBDB-T devices: PBDB-T was dissolved in CF solution (10 mg ml⁻¹) and spin-coated on ZnO (3000 rpm) to form a 70 nm thick film. Neat Y6 devices: Y6 was dissolved in CF solution (16 mg ml⁻¹) and spin-coated on ZnO (3000 rpm) to form a 70 nm thick film.

**EQE measurements.** A homebuilt setup including a Perkin Elmer UV/VIS/NIR spectrometer (LAMBDA 950) as a source for monochromatic light was used. The light was chopped at 273 Hz and directed onto the device under test (DUT). The resulting photocurrent was amplified by a low noise current amplifier (FEMTO DLPCA-200) and measured with the Stanford SR860 lock-in amplifier. To decrease

the noise floor of the setup, the DUT was mounted in an electrically shielded and temperature controlled Linkam sample stage. An integration time up to 1000 s was used for detecting wavelengths above 1500 nm. NIST-calibrated silicon and Ge photodiodes from Newport were used as a calibration reference. The temperature inside the Linkam sample stage was set to −120 to 60 °C by the Linkam T96 temperature controller in combination with an LNP96 liquid nitrogen pump. The commercial amorphous silicon thin film solar cell, used for temperature dependent EQE measurements, was manufactured by TRONY with the part number sc80125s-8.

**Gaussian fits.** The EQE associated with CT states and mid-gap states were routinely fitted in accordance with the standard Marcus charge-transfer formalism. The associated EQEs are given by $\mathrm{EQE}_{CT}(E) = g(E, E_{CT}, \lambda_{CT}, f_{CT})$ and $\mathrm{EQE}_t(E) = g(E, E_t, \lambda_t, f_t)$ for CT and mid-gap state absorption, respectively, where

$$g\left(E, E_j, \lambda_j, f_j\right) = f_j E^{-1}\left(4\pi\lambda_j kT\right)^{-\frac{1}{2}}\exp\left(-\frac{\left(E_j + \lambda_j - E\right)^2}{4\lambda_j kT}\right). \quad (5)$$

Here, $f_j$, $E_j$ and $\lambda_j$ are fitting parameters. All fit parameters obtained in this work are summarized in Supplementary Table 1. The EQEs are assumed to be related to their respective absorption coefficients as $\alpha_{CT}(E) \propto \mathrm{EQE}_{CT}(E)$ and $\alpha_t(E) \propto \mathrm{EQE}_t(E)$ (neglecting interference effects).

## Data availability
The data that support the findings of this study are available from the corresponding author upon reasonable request.

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

## Acknowledgements

This work was funded through the Welsh Government's Sêr Cymru II Program 'Sustainable Advanced Materials' (Welsh European Funding Office — European Regional Development Fund). C.K. is recipient of a UKRI EPSRC Doctoral Training Program studentship. N.Z. is funded by a studentship through the Sêr Cymru II Program. P.M. is a Sêr Cymru II Research Chair and A.A. is a Rising Star Fellow also funded through the Welsh Government's Sêr Cymru II 'Sustainable Advanced Materials' Program (European Regional Development Fund, Welsh European Funding Office and Swansea University Strategic Initiative). This work was also funded by UKRI through the EPSRC Program Grant EP/T028511/1 Application Targeted Integrated Photovoltaics.

## Author contributions

A.A. and P.M. provided the overall leadership of the project. O.J.S. and A.A. conceptualized the idea. C.K. and A.A. designed the experiments. C.K. performed most measurements, analyzed the data, and performed optical modeling. OJS developed the theoretical model. C.K., O.J.S. and A.A. interpreted the data. N.Z. assisted with EQE$_{PV}$ measurements. W.L. fabricated the devices. All co-authors contributed in the development of the manuscript which was initially drafted by C.K.

## Competing interests

The authors declare no competing interests.
