## [Peer Review File · Nature Communications]

REVIEWERS' COMMENTS

Reviewer #1 (Remarks to the Author):

Subgap absorption in polymer solar cell D-A blend system has become a highly interesting issue because it contains the information of CT state as well as disorders presented in the materials. Through device measurements on three types of systems ranging from conventional fullerene acceptor to the novel non-fullerene acceptor, the authors applied a number of models to simulate the absorption tails and concluded that the near onset subgap absorption is due to the static disorder of Gaussian type, and the lower energy part is due to the thermal broadening. This finding can clarify the previous view on Urbach energy, a concept borrowed from inorganic semiconductor.

I find the corrections are satisfactory. I recommend the publication in the present form.

Reviewer #2 (Remarks to the Author):

I'm happy with the responses to my questions and concerns, and would like to compliment the authors with the adequate changes and additions to the manuscript. The work can in my view be published as is.

A very minor point: somehow the resolution of Fig. S10 is rather low.

Reviewer #3 (Remarks to the Author):

The paper investigates the causes to the lineshape of the absorption at and below the absorption edge in disordered organic semiconductors. The authors show that this is composed of a Gaussian part, attributed to static disorder, and an exponential part, assigned to an Urbach tail with Urbach energy of kT . This is an important and general finding that should be published in *Nature Communications*.

There is only one aspect where the study has some weakness which was already raised by a previous reviewer and which the authors have only partially addressed. The authors use the EQE of solar cells as a proxy for the absorption spectra due to the high sensitivity of EQE measurements. The EQE is the product of absorption times exciton dissociation times charge extraction. Their approach assumes that exciton dissociation and charge extraction are independent of the incident photon energy. I fully agree with the authors that this is valid for efficient solar cells made from blends, but it becomes more debatable when neat donor and acceptor films are investigated, where dissociation is often extrinsic at traps, defects or interfaces to the electrodes. The authors have now stated their underlying assumptions, and that is a valid approach (though repeating this caveat when it comes to the neat film would not harm), and the paper can, in principle, go ahead as it is. Nevertheless, I want to emphasize that the convincing power of the paper would gain substantially, if the authors were to also analyse at least one or two actual absorption spectra, e.g. taken by photothermal deflection spectroscopy (PDS). This could be simply literature data, and an inclusion of such data in the SI would be fully sufficient to swipe away any possible concerns about the approach. I have not done an extensive literature research, yet with a quick superficial check found the PDS spectrum of neat PC60BM (Fig. 2b in *JACS*, 2015, 137, 5256). Is this consistent with the data presented on neat PC70BM in Fig. 2b of the submitted paper?

It is up to the authors to which extent they want to follow up these comments and ideas. The submitted manuscript is, in any case, an important and stimulating contribution with a valid approach (as the assumptions are made clear), and I recommend publication in *Nature Communications*.

Response Letter

Changes made to the revised manuscript and Supplementary Information have been indicated in **Red**.

Reviewer #1 (Remarks to the Author):

Subgap absorption in polymer solar cell D-A blend system has become a highly interesting issue because it contains the information of CT state as well as disorders presented in the materials. Through device measurements on three types of systems ranging from conventional fullerene acceptor to the novel non-fullerene acceptor, the authors applied a number of models to simulate the absorption tails and concluded that the near onset subgap absorption is due to the static disorder of Gaussian type, and the lower energy part is due to the thermal broadening. This finding can clarify the previous view on Urbach energy, a concept borrowed from inorganic semiconductor. I find the corrections are satisfactory. I recommend the publication in the present form.

Answer: We would like to thank the reviewer for their positive feedback on our work.

Reviewer #2 (Remarks to the Author):

I'm happy with the responses to my questions and concerns, and would like to compliment the authors with the adequate changes and additions to the manuscript. The work can in my view be published as is. A very minor point: somehow the resolution of Fig. S10 is rather low.

Answer: We thank reviewer for complimenting on the substantial changes we made to the manuscript. The resolution of Supplementary Fig. 11 (former Fig. S10) is improved in the new version of the Supplementary Information.

Reviewer #3 (Remarks to the Author):

The paper investigates the causes to the lineshape of the absorption at and below the absorption edge in disordered organic semiconductors. The authors show that this is composed of a Gaussian part, attributed to static disorder, and an exponential part, assigned to an Urbach tail with Urbach energy of kT . This is an important and general finding that should be published in Nature Communications. There is only one aspect where the study has some weakness which was already raised by a previous reviewer and which the authors have only partially addressed. The authors use the EQE of solar cells as a proxy for the absorption spectra due to the high sensitivity of EQE measurements. The EQE is the product of absorption times exciton dissociation times charge extraction. Their approach assumes that exciton dissociation and charge extraction are independent of the incident photon energy. I fully agree with the authors that this is valid for efficient solar cells made from blends, but it becomes more debatable when neat donor and acceptor films are investigated, where dissociation is often extrinsic at traps, defects or interfaces to the electrodes. The authors have now stated their underlying

assumptions, and that is a valid approach (though repeating this caveat when it comes to the neat film would not harm), and the paper can, in principle, go ahead as it is. Nevertheless, I want to emphasize that the convincing power of the paper would gain substantially, if the authors were to also analyse at least one or two actual absorption spectra, e.g. taken by photothermal deflection spectroscopy (PDS). This could be simply literature data, and an inclusion of such data in the SI would be fully sufficient to swipe away any possible concerns about the approach. I have not done an extensive literature research, yet with a quick superficial check found the PDS spectrum of neat PC60BM (Fig. 2b in JACS, 2015, 137, 5256). Is this consistent with the data presented on neat PC70BM in Fig. 2b of the submitted paper? It is up to the authors to which extent they want to follow up these comments and ideas. The submitted manuscript is, in any case, an important and stimulating contribution with a valid approach (as the assumptions are made clear), and I recommend publication in Nature Communications.

Answer: We would like to thank the reviewer for their very valuable suggestion and positive feedback. In light of this comment, we have added Supplementary Fig. 3 showing that $E_{\text{U}}^{\text{app}} \approx kT$ is reobtained from the PDS spectrum of neat PC₆₀BM as taken from the Fig. 2b in the reference JACS, 2015, 137, 5256.

Changes the main text:

“We note that this is consistent with photothermal deflection spectroscopy results³³ of neat PC₆₀BM (see Supplementary Fig. 3), confirming the underlying assumption that the spectral line shape of the EQE follows α in the sub-gap. In PC₇₀BM, the spectral range where $E_{\text{U}}^{\text{app}} \approx kT$ is limited by deep trap-state absorption at low energies, and low signal-to-noise ratio (SNR) due to the poor exciton dissociation in the neat phase (translating into a low IQE). However, adding 0.1 mol% of the wide-gap donor m-MTDATA results in enhanced exciton dissociation, which improves SNR and increases the spectral range within which $E_{\text{U}}^{\text{app}} \approx kT$ can be observed.”

Changes in the Supplementary Information:

Supplementary Figure 3. a, Sub-gap EQE of a PC₇₀BM solar cell (black line) and the absorption coefficient (α ; green line) of a neat PC₆₀BM film measured via photothermal deflection spectroscopy (PDS) are compared. The PDS data was taken from the literature.⁹ **b**, The corresponding E_U^{app} spectra calculated from the EQE of a PC₇₀BM solar cell (black line) and from the α of a neat PC₆₀BM film (green line) are shown. The EQE and PDS derived E_U^{app} spectra show $E_U^{\text{app}} \approx kT$ at similar energies at around 1.70 eV and 1.65 eV, respectively, confirming that the spectral line-shape of EQE can be used as an approximation for α in the sub-gap energy range.

In the following we have listed all major changes made in the main manuscript and Supplementary Information apart from the changes already mentioned as part of the response to reviewer three.

- Since we added Supplementary Fig. 3, the labels and references of the Figures in the main text and Supplementary Information were revised accordingly (marked in red in the main text).
- We amended the method section, because the device fabrication of the neat Y6 device has been missing so far.

“Y6 devices: Y6 was dissolved in chloroform solution (16 mg ml⁻¹) and spin-coated on ZnO (3000 rpm) to form a 70 nm thick film.”

- We have substituted the word absorbance with the word absorptance in the main text in order to use correct terminology.

“The spectral line-shape of α and the absorptance A in the sub-gap tail are generally related via a modified Beer-Lambert law, $A = \tilde{f}\alpha d$, where d is the thickness of the active layer and \tilde{f} is an energy-dependent correction factor accounting for optical interference.^{27,28}”